# t^6^A and ms^2^t^6^A Modified Nucleosides in Serum and Urine as Strong Candidate Biomarkers of COVID-19 Infection and Severity

**DOI:** 10.3390/biom12091233

**Published:** 2022-09-03

**Authors:** Yu Nagayoshi, Kayo Nishiguchi, Ryosuke Yamamura, Takeshi Chujo, Hiroyuki Oshiumi, Hiroko Nagata, Hitomi Kaneko, Keiichi Yamamoto, Hirotomo Nakata, Korin Sakakida, Akihiro Kunisawa, Masataka Adachi, Yutaka Kakizoe, Takanori Mizobe, Jun-ichi Kuratsu, Shinya Shimada, Yasushi Nakamori, Masao Matsuoka, Masashi Mukoyama, Fan-Yan Wei, Kazuhito Tomizawa

**Affiliations:** 1Department of Molecular Physiology, Faculty of Life Sciences, Kumamoto University, Kumamoto 860-8556, Japan; 2Department of Nephrology, Faculty of Life Sciences, Kumamoto University, Kumamoto 860-8556, Japan; 3Department of Immunology, Faculty of Life Sciences, Kumamoto University, Kumamoto 860-8556, Japan; 4Department of Laboratory Medicine, Kumamoto University Hospital, Kumamoto 860-8556, Japan; 5Department of Hematology, Rheumatology and Infectious Diseases, Faculty of Life Sciences, Kumamoto University, Kumamoto 860-8556, Japan; 6Department of Metabolic Medicine, Faculty of Life Sciences, Kumamoto University, Kumamoto 860-8556, Japan; 7Shimadzu Corporation, Kyoto 604-8442, Japan; 8Kumamoto Kenhoku Hospital, Kumamoto 865-8005, Japan; 9Sakurajyuji Hospital, Kumamoto 861-4173, Japan; 10JCHO Kumamoto General Hospital, Kumamoto 866-8660, Japan; 11Department of Emergency and Critical Care Medicine, Kansai Medical University General Medical Center, Osaka 570-8507, Japan; 12Department of Modomics Biology and Medicine, Institute of Development Aging and Cancer, Tohoku University, Sendai 980-8575, Japan

**Keywords:** COVID-19, modified nucleosides, LC-MS

## Abstract

SARS-CoV-2 infection alters cellular RNA content. Cellular RNAs are chemically modified and eventually degraded, depositing modified nucleosides into extracellular fluids such as serum and urine. Here we searched for COVID-19-specific changes in modified nucleoside levels contained in serum and urine of 308 COVID-19 patients using liquid chromatography-mass spectrometry (LC-MS). We found that two modified nucleosides, *N*^6^-threonylcarbamoyladenosine (t^6^A) and 2-methylthio-*N*^6^-threonylcarbamoyladenosine (ms^2^t^6^A), were elevated in serum and urine of COVID-19 patients. Moreover, these levels were associated with symptom severity and decreased upon recovery from COVID-19. In addition, the elevation of similarly modified nucleosides was observed regardless of COVID-19 variants. These findings illuminate specific modified RNA nucleosides in the extracellular fluids as biomarkers for COVID-19 infection and severity.

## 1. Introduction

Coronavirus disease 2019 (COVID-19) is a respiratory infectious disease caused by the severe acute respiratory syndrome coronavirus 2 (SARS-CoV-2) [1,2]. This disease spread quickly around the world, causing millions of deaths. For confirmed diagnosis of COVID-19 at the bedsides, RT-PCR test targeting viral genome RNA and antigen test against viral spike proteins are mainly used. However, there are multiple problems in these clinical examinations. One major problem is that these tests show only negative or positive results. Therefore, they are not suitable for determining or predicting the severity of this disease. Some serum proteins such as CCL17 and IFN-gamma3 were reported as the biomarker for COVID-19 severity, but the specificities of these biomarkers are not high because of elevation in other diseases [3,4]. Another problem is the risk of infection from clinical samples. Currently, RNA extracted from saliva or pharyngeal swabs is used in both tests, and handling the samples always exposes healthcare workers to infection risks. SARS-CoV-2 is undetectable in serum and urine [5,6]. Therefore, if an appropriate diagnosis method is devised, blood and urine are ideal samples for COVID-19 diagnosis.

SARS-CoV-2 is an RNA virus, having a single-stranded, ~30 kb-long RNA genome with 12 open reading frames (ORFs) [7]. In ORF1a and 1b, two RNA modification enzymes are encoded. One is guanine *N*^7^-methyltransferase catalyzing 5′ terminal cap modification which prevents recognition by the host immunity and promotes SARS-CoV-2 protein synthesis [8]. The other is 2′-*O*-methyltransferase whose modification also contributes to the formation of cap structure and suppresses recognition by the host innate immune system [8]. Moreover, highly modified regions were suggested to exist in SARS-CoV-2 genome RNA and its transcripts using nanopore direct RNA sequencing [7]. These reports suggest that RNA modifications in SARS-CoV-2 play important roles in viral replication and self-defense. However, the clinical implications are completely unclear.

Over 100 kinds of chemical modifications of RNA are reported in the three domains of life and they have a variety of biochemical functions [9]. For example, *N*^6^-threonylcarbamoyladenosine (t^6^A) modification exists at position 37 of tRNAs that decipher ANN codons, and t^6^A governs the accuracy and efficiency of protein synthesis in the cytosol [10]. t^6^A modification is introduced by a protein complex called the kinase, putative endopeptidase, and other proteins of small size (KEOPS) [11]. Due to the physiological importance of t^6^A, the deficits of KEOPS components cause nephrotic syndrome and primary microcephaly [12]. t^6^A in tRNA^Lys^
_UUU_ is further thiomethylated by CDKAL1 protein, resulting in 2-methylthio-*N*^6^-threonylcarbamoyladenosine (ms^2^t^6^A) modification [13,14]. ms^2^t^6^A modification is important for proinsulin synthesis, and the deficit of this modification causes the development of type 2 diabetes [14]. Due to the physiological importance of tRNA modifications, the deficits of various other tRNA modifications in mammals also cause various diseases including mitochondrial diseases and neurological disorders [15,16,17]. At the end of its life, modified RNA is degraded into single nucleosides, and modified nucleosides are excreted to extracellular spaces, circulated in serum, and discarded into the urine [18,19].

In this study, we have identified a characteristic elevation in specific modified nucleosides through infection experiments on cultured cells. These modified nucleosides were significantly elevated in serum and urines of COVID-19 patients and might be useful for novel biomarkers of COVID-19.

## 2. Materials and Methods

### 2.1. Cell Culture and Viral Infection

ACE2-overexpressing HEK293 cells were maintained in DMEM (low glucose) with 10% heat-inactivated fetal calf serum (FCS) and penicillin-streptomycin solution (P/S). SARS-CoV-2 JPN/TY/WK-521 strain was obtained from the National Institute of Infectious Diseases in Japan and amplified with VeroE6/TMPRSS2 cells. ACE2-overexpressing HEK293 cells were infected by SARS-CoV-2 particles at an MOI 1.0. RNA extraction by TRIZOL Reagent (Thermo Scientific, Waltham, MA, USA) was performed 18 h after the infection. The extracted RNAs were degraded into single nucleosides using nuclease P1 and alkaline phosphatase.

### 2.2. Sample Preparation and LC-MS Analysis of Modified RNA Nucleosides

Nucleosides from culture cells were desalted at 4 °C, 12,000 rpm, 30 min centrifugation with Nanosep 3K Omega (Pall Corporation, New York, USA). Modified nucleoside quantification was performed by a triple quadrupole mass spectrometry system (LCMS-8050, Shimadzu Corporation, Kyoto, Japan) equipped with an electrospray ionization (ESI) source and an ultra-high performance liquid chromatography system [19]. The nucleoside samples were injected into an Inertsil ODS-3 column (GL Science, Tokyo, Japan). The mobile phase consisted of two types of solutions. One is 5 mM ammonium acetate in water adjusted to pH 5.3, and the other is 60% (*v*/*v*) acetonitrile in water. The LC gradient was set as follows: 1–10 min: 1–22.1% B, 10–15 min: 22.1–63.1% B, 15–17 min: 63.1–100% B, 17–22 min: 100% B, and 22–23 min, 100–0.6% B. The flow rate was 0.4 mL/min, and the injection volume was 2 μL. Detection was performed in the MRM (multiple reaction monitoring) modes of LabSolutions System (Shimadzu Corporation). The MRM transitions for modified nucleosides in this method are described in Appendix A. Interface temperature was 300 °C, desolvation line temperature was 250 °C, and heat block temperature was 400 °C. Nitrogen gas was supplied from an N2 feeder Model T24FD (System Instruments, Tokyo, Japan) for nebulization and drying, and argon gas was used for collision-induced dissociation.

### 2.3. Automatic Sample Preparation and LC-MS Analysis for t^6^A and ms^2^t^6^A in Serum and Urine Samples

Serum and urine samples were desalted and deproteinized by a fully automated sample preparation module (CLAM-2030, Shimadzu Corporation) coupled to an LCMS-8050. Twenty microliters of the sample was automatically delivered to a polytetrafluoroethylene filter vial (0.45 μm pore size) which was pre-conditioned with 20 μL methanol. Eighty microliters of methanol and 20 μL of isopropanol was added to the filter vial and stirred for 60 s. The samples were filtrated and delivered to LC-MS/MS system with 20 μL water. t^6^A and ms^2^t^6^A quantification was performed by the same LCMS-8050 system described above. The serum samples were injected into a Mastro2 C18 column (Shimadzu GLC Ltd., Tokyo, Japan) from CLAM-2030 automatically. The mobile phase consisted of two types of solutions. One is 0.1% (*v*/*v*) formic acid in water (A), and the other is 0.1% (*v*/*v*) formic acid in acetonitrile (B). The LC gradient was set as follows: 1–1.2 min 10–20% B, 1.2–3.8 min 20–35% B, 3.8–4.5 min 35–90% B, 4.5–4.7 min 90–10% B. The flow rate was 0.3 mL/min and the injection volume was 2 μL. Detection was performed in the MRM (multiple reaction monitoring) modes of LabSolutionsSystem (Shimadzu Corporation, Kyoto, Japan). The MRM transitions for modified nucleosides in this method are described in Appendix A. Interface temperature was 270 °C, desolvation line temperature was 250 °C, and heat block temperature was 400 °C. Nitrogen gas was supplied from an N2 feeder Model T24FD for nebulization and drying, and argon gas was used for collision-induced dissociation.

### 2.4. Patients and Severity Assessment

We enrolled COVID-19 patients diagnosed by real-time reverse transcription-polymerase chain reaction (RT-PCR) using extracted RNAs from saliva or pharyngeal swabs (Table 1). The presence of mutations was examined using TaqMan SARS-CoV-2 Mutation Panel (Thermo Scientific, Waltham, MA, USA). The severity definitions of COVID-19 were based on the Spectrum of SARS-CoV-2 Infection from the “COVID-19 Treatment Guidelines” of NIH. We classified COVID-19 patients into two groups: asymptomatic/mild and moderate/severe. Moderate patients were classified as having pneumonia and requiring oxygen administration, and severe patients as requiring ventilator management and extracorporeal circulation. The mild patients had various signs and symptoms of COVID-19, for example, fever, cough, sore throat, malaise, headache, muscle pain, nausea, vomiting, diarrhea, loss of taste and smell but who do not have shortness of breath, dyspnea, or abnormal chest imaging by CT scan. Asymptomatic patients had no symptoms of COVID-19. Patients with other infectious diseases, including bacterial pneumonia and other viral infection, were diagnosed by clinical investigators with various examinations including blood culture tests, pneumococcal urinary antigen tests, and flu tests performed before the COVID-19 pandemic. The information from these patients is described in Appendix A. We collected serum from the same COVID-19 patients at the infection period and recovery period. A recovery period was defined by the resolution of fever and other symptoms.

### 2.5. Statistical Analysis

Data accorded with normal distribution and homogeneity of variance were expressed as the mean ± standard error of means (S.E.M) and compared by Mann–Whitney U tests. Categorical variables were compared by the Kruskal–Wallis test and Dunn’s multiple comparison tests. For calculation of sensitivity and specificity, we used receiver operating characteristic analysis to discriminate between healthy volunteers and COVID-19 patients. Statistical analyses were performed with the Prism 9 software (GraphPad, San Diego, CA, USA), and a *p*-value less than 0.05 was considered statistically significant.

## 3. Results

To identify modified nucleosides whose amount specifically changes in COVID-19, we first performed an infection experiment using angiotensin converting enzyme 2 (ACE2)-overexpressing human embryonic kidney (HEK) 293 cells. SARS-CoV-2 particles were infected at an MOI of 1. After 18 h of incubation, we extracted total RNA and degraded it into single nucleosides using nuclease P1 and alkaline phosphatase. We then quantified modified nucleosides by LC-MS. As a result, within the total RNA of SARS-CoV-2-infected cells, we observed elevation of six modified nucleosides, which are N^1^-methyladenosine (m^1^A), *N*^2^,*N*^2^-dimethylguanosine (m^2^_2_G), *N*^6^-threonylcarbamoyladenosine (t^6^A), 2-methylthio-*N*^6^-threonylcarbamoyladenosine (ms^2^t^6^A), *N*^6^-methyl-*N*^6^-threonylcarbamoyladenosine (m^6^t^6^A), and *N*^6^,2′-O-dimethyladenosine (m^6^Am) (Figure 1a). Especially, t^6^A and ms^2^t^6^A (Figure 1b) were over 4 times elevated compared to control cells. From this result, t^6^A and ms^2^t^6^A were judged as good candidate biomarkers for SARS-CoV-2 infection.

Next, to investigate if t^6^A and ms^2^t^6^A within human urine can be used as SARS-CoV-2 infection biomarkers, we performed LC-MS analysis using the urine of patients with COVID-19. These patients were diagnosed by RT-PCR test against SARS-CoV-2 genome RNA from saliva or pharyngeal swabs (Table 1). Urine is highly susceptible to physiological conditions, and appropriate normalization is essential for the urine test. Generally, urine creatinine is the most commonly used normalization substance. Therefore, we analyzed t^6^A and ms^2^t^6^A in urine normalized by urine creatinine and these results were compared to healthy samples (Figure 2a,b). The t^6^A and ms^2^t^6^A levels in urine were significantly increased in COVID-19 patients. We also performed receiver-operating characteristic (ROC) analysis using data of t^6^A and ms^2^t^6^A normalized by urine creatinine. On t^6^A, setting the cutoff value to 344,420 resulted in a sensitivity of 71.7%, a specificity of 77.8%, and a likelihood ratio of 3.23 (Figure 2c). Regarding ms^2^t^6^A, setting the cutoff value to 76,878 resulted in a sensitivity of 86.6%, a specificity of 91.7%, and a likelihood ratio of 2.6 (Figure 2d).

To investigate if elevations of t^6^A and ms^2^t^6^A in urine are characteristic of COVID-19, we also compared the patient urine of COVID-19 with other infectious diseases including influenza and bacterial pneumonia. The elevation of t^6^A and ms^2^t^6^A in urine was observed only in the COVID-19 group (Figure 3a,b). From these results, measurements of t^6^A and ms^2^t^6^A in urine were observed to have the equivalent diagnostic ability to the RT-PCR test for COVID-19.

Next, to investigate if t^6^A and ms^2^t^6^A within human serum can be used as SARS-CoV-2 infection biomarkers, we measured t^6^A and ms^2^t^6^A in serum normalized by unmodified adenosine and compared them with healthy samples. The t^6^A and ms^2^t^6^A levels were significantly elevated in the serum of COVID-19 patients (Figure 4a,b). We also performed ROC analysis using data of t^6^A and ms^2^t^6^A. On t^6^A, setting the cutoff value to 1.039 resulted in a sensitivity of 98.4%, a specificity of 92.5%, and a likelihood ratio of 13.12 (Figure 4c). Regarding ms^2^t^6^A, setting the cutoff value to 0.1034 resulted in a sensitivity of 94.2%, a specificity of 92.5%, and a likelihood ratio of 12.55 (Figure 4d).

Next, to investigate if t^6^A and ms^2^t^6^A within human serum can be used as quantitative biomarkers to determine the severity of SARS-CoV-2 infection, we first examined the patients’ conditions from medical records and classified them by severity. Based on the Clinical Spectrum of SARS-CoV-2 Infection from “COVID-19 Treatment Guidelines” of NIH, we classified COVID-19 patients into two groups: asymptomatic/mild and moderate/severe. Then, we compared the measurements of t^6^A and ms^2^t^6^A in the serum of these two groups against a healthy group. As a result, as the severity of COVID-19 worsened, ms^2^t^6^A in serum also increased (Figure 5a,b). Next, we confirmed the relationships between the measurements t^6^A and ms^2^t^6^A in serum and clinical indicators related to COVID-19 severity (Table 1). Our results show the levels of lactate dehydrogenase (LDH), C-reactive proteins (CRP), and lymphocyte percentage in COVID-19 patients significantly correlated with t^6^A and ms^2^t^6^A levels (Appendix A).

We also compared the changes in serum t^6^A and ms^2^t^6^A levels within the same COVID-19 moderate/severe patients at the infection period and recovered period. We found that t^6^A and ms^2^t^6^A in serum significantly decreased at the recovered period (Figure 5c,d). Based on these results, the measurement of t^6^A and ms^2^t^6^A in serum could be useful to determine the severity and the effect of treatment.

Since the end of 2020, patients with variants of SARS-CoV-2 have been reported from various regions, including the United Kingdom (B1.1.7), South Africa (B1.351), Brazil (P1), and India (B.1.617.2, AY.1, AY.2) [20]. These variants are often associated with enhanced transmissibility and evasion from host antibodies. We collected the serum of patients with B1.1.7 (α) and B.1.617.2 (δ) variants of SARS-CoV-2. Using the same LC-MS method, we measured t^6^A and ms^2^t^6^A in the serum of patients infected with these variants, and we found that t^6^A and ms^2^t^6^A were also elevated in the serum of patients infected with all monitored variants (Figure 6a,b). These results suggest that the diagnosis of COVID-19 by measuring t^6^A and ms^2^t^6^A in serum could be useful regardless of variants of SARS-CoV-2 spike protein.

## 4. Discussion

In this study, we first found characteristic elevations of specific modified nucleosides t^6^A and ms^2^t^6^A during SARS-CoV-2 infection experiments. These biomolecules were also elevated in the serum and urine of COVID-19 patients. Moreover, these elevations correlated with the severity and recovery of COVID-19. In the serum of patients infected with several mutant strains, these elevations were also observed.

To examine the presence of SARS-CoV-2, RT-PCR tests and antigen tests are easy and useful. However, clinical samples for these tests, which are saliva and nasopharyngeal swabs, often contain SARS-CoV-2, constantly exposing healthcare workers to the risk of infections during the collection and handling of these samples. Serum and urine contain very little of the SARS-CoV-2 virion [5,6]. Therefore, the establishment of COVID-19 diagnosis using modified nucleosides in serum and urine could provide more safety and less stress for healthcare workers. Considering the inaccessibility of mass spec machines in many facilities, we are currently trying to develop an easy and inexpensive t^6^A ELISA kit for COVID-19 detection using safer serum or urine samples rather than dangerous saliva and pharyngeal swabs.

In COVID-19 treatment, RT-PCR tests and antigen tests are not suitable for the proper assessment of COVID-19 severity. PCR tests and antigen tests for the SARS-CoV-2 viral genome from saliva or nasopharyngeal swab have no correlation with COVID-19 severity [21,22,23,24]. From our study, the elevations of t^6^A and ms^2^t^6^A in serum correlated with the severity and recovery of infection. The measurements of t^6^A and ms^2^t^6^A in serum could contribute to the appropriate assessment of severity and treatment effect, as well as to appropriately evaluate the efficacy of therapeutic agents during clinical trials. In this study, we examine only the elevation of t^6^A and ms^2^t^6^A in serum of patients with α and δ variants, and elevations in serum by infections with other variants should be checked.

From our study, the sources of t^6^A and ms^2^t^6^A are unclear, although there are some candidates. One is the result of cell damage to immune cells and/or tissue cells upon infection. When the host is infected by pathogens, large numbers of tissue cells and immune cells react and finally collapse. Our in vitro data using HEK293 cells indicate these elevations of modified nucleosides may be related to tissue cell damage. Moreover, we found that these elevations of modified nucleosides in serum correlated with LDH (Appendix A). Upon destruction of tissue or immune cells, many modified nucleosides leak into the extracellular region and where they accumulate [18,19]. Therefore, the correlations of serum t^6^A and ms^2^t^6^A with COVID-19 severity might reflect the damage of tissue and/or immune cells upon SARS-CoV-2 infection. Another potential source of t^6^A and ms^2^t^6^A is the genome RNA of SARS-CoV-2. Within the viral RNA, chemically modified regions were detected using nanopore sequencing experiments, although the modification species are unidentified [7]. No obvious candidates for enzymes that modify t^6^A and ms^2^t^6^A are encoded in the genome RNA of SARS-CoV-2. Therefore, if the viral RNA contains t^6^A and ms^2^t^6^A, SARS-CoV-2 likely uses the host’s modifying enzymes, the KEOPS complex for t^6^A modification and CDKAL1 for ms^2^ modification [11,14,25]. Some RNA viruses, such as HIV-1, have been reported to use the host RNA modification enzyme to escape from host immunity [26]. In future studies, it will be necessary to monitor changes in the expression of these modifying enzymes upon viral infection, as well as the modification levels of the host tRNAs and other RNAs. Recently, many types of vaccinations, including mRNA vaccines, were certified and used in many countries to combat the COVID-19 pandemic. It will be important to investigate the changes of t^6^A and ms^2^t^6^A in vaccinated patient serum in future studies.

## 5. Conclusions

In summary, we discovered serum and urine t^6^A and ms^2^t^6^A nucleosides as effective biomarkers of COVID-19. Modified nucleosides are conceptually new metabolites to be measured in the clinical area, and our study is the first to monitor them in COVID-19. The most important merits of the modified nucleoside test over the RT-PCR test are: (1) correlation of serum t^6^A and ms^2^t^6^A levels with the severity and recovery and (2) accuracy of this test regardless of the mutation in the spike protein of SARS-CoV-2. This test is the first evidence for diagnosis using modified nucleosides for COVID-19 and could be useful for accurate assessment of COVID-19 severity and recovery.

## Figures and Tables

**Figure 1 biomolecules-12-01233-f001:**
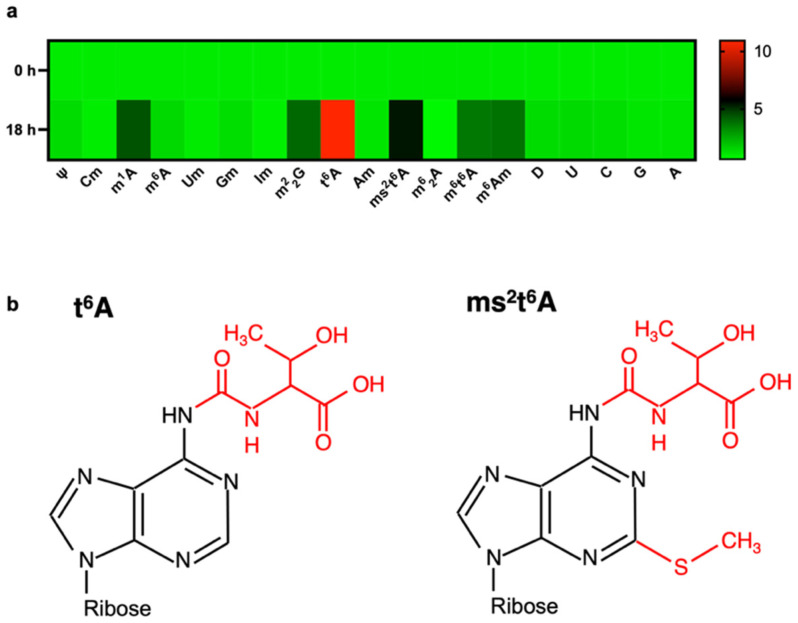
Screening of candidate modified nucleosides by SARS-CoV-2. (**a**) Heatmap analysis of modified nucleoside levels in extracted RNA from SARS-CoV-2-infected HEK293 cells. Color scale shows the auto-scaled relative mean of *n* = 3 biological replicates. ψ: pseudouridine; Cm: 2′-O-methylcytidine; m^1^A: *N*^1^-methyladenosine; m^6^A: *N*^6^-methyladenosine; Um: 2′-O-methyluridine; Gm: 2′-O-methylguanosine; Im: 2′-O-methylinosine; m^2^_2_G: *N*^2^,*N*^2^-dimethylguanosine; t^6^A: *N*^6^-threonylcarbamoyladenosine; Am: 2′-O-methyladenosine; ms^2^t^6^A: 2-methylthio-*N*^6^-threonylcarbamoyladenosine; m^6^,_2_A: *N*^6^,*N*^6^-dimethyladenosine; m^6^t^6^A: *N*^6^-methyl-threonylcarbamoyladenosine; m^6^Am: *N*^6^,2′-O-dimethyladenosine; D: dihydrouridine; U: uridine; C: cytidine; G: guanosine; A: adenosine. (**b**) Chemical structures of t^6^A and ms^2^t^6^A. Modified residues are depicted in red.

**Figure 2 biomolecules-12-01233-f002:**
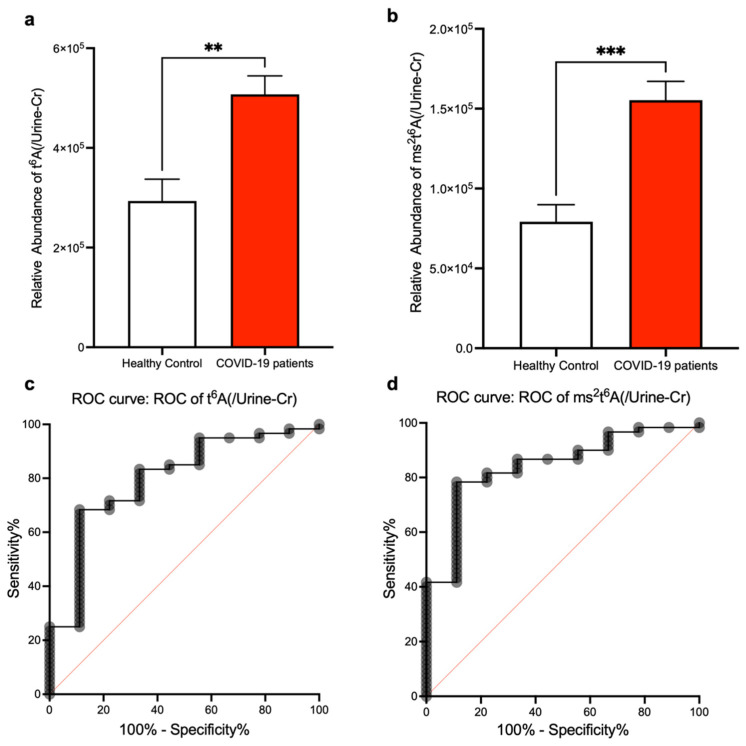
SARS-CoV-2 infection causes elevation of t^6^A and ms^2^t^6^A levels in urine. (**a**,**b**) Measurements of t^6^A (**a**) and ms^2^t^6^A (**b**) in urine and comparison with healthy volunteers. LC-MS peak areas of t^6^A and ms^2^t^6^A divided by urine creatinine levels are shown. ** *p* < 0.01, *** *p* < 0.001 by Mann–Whitney U test. (**c**,**d**) ROC analysis for measurements of t^6^A (**c**) and ms^2^t^6^A (**d**) in urine was performed for calculation of sensitivity and specificity.

**Figure 3 biomolecules-12-01233-f003:**
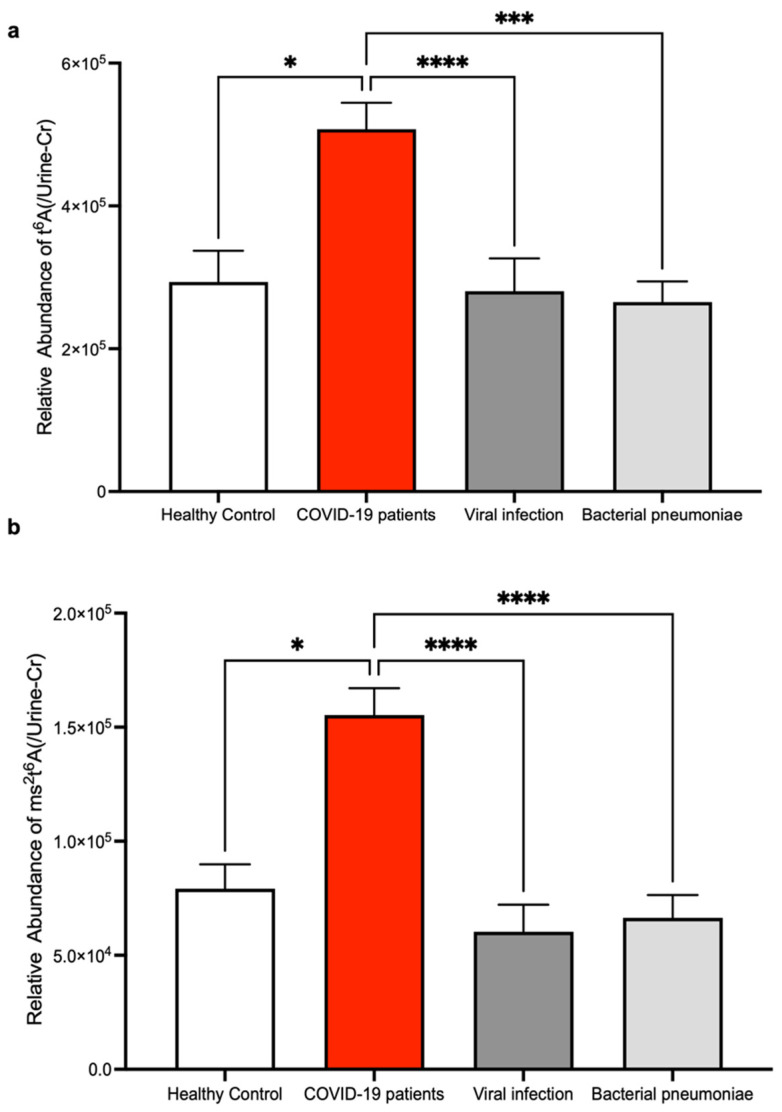
The comparison of t^6^A and ms^2^t^6^A levels in the urine of COVID-19 patients with other infectious diseases. (**a**,**b**) Comparison of t^6^A (**a**) and ms^2^t^6^A (**b**) levels in the urine of patients with viral infection or bacterial pneumoniae. LC-MS peak areas of t^6^A and ms^2^t^6^A divided by urine creatinine levels are shown. * *p* < 0.05, *** *p* < 0.001, **** *p* < 0.0001 by Kruskal–Wallis test and Dunn’s multiple comparison test.

**Figure 4 biomolecules-12-01233-f004:**
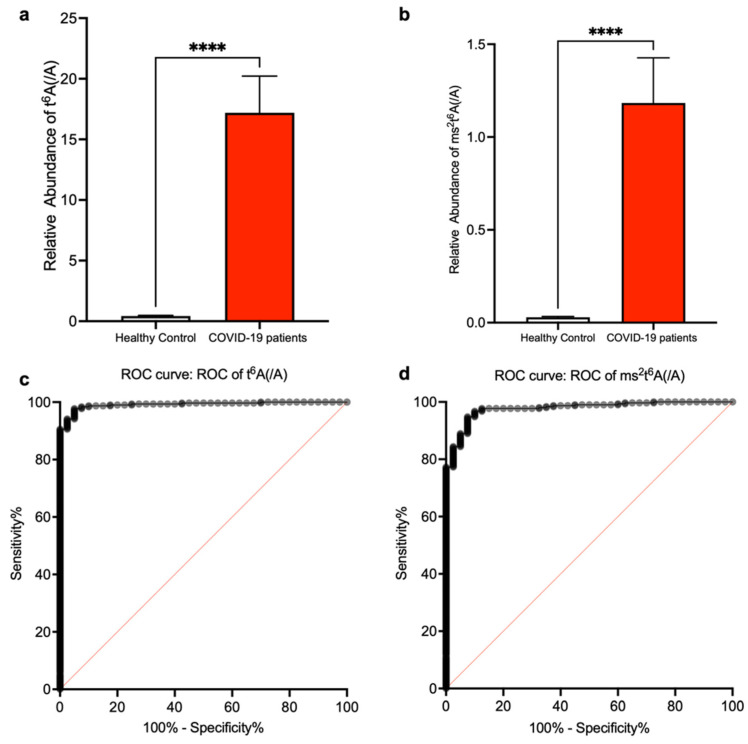
Elevation of t^6^A and ms^2^t^6^A in serum of COVID-19 patients. (**a**,**b**) Measurements of t^6^A (**a**) and ms^2^t^6^A (**b**) in serum of COVID-19 patients and comparison with healthy volunteers. LC-MS peak areas of t^6^A or ms^2^t^6^A divided by LC-MS peak areas of adenosine are shown. **** *p* < 0.0001 by Mann–Whitney U test. (**c**,**d**) ROC analysis for t^6^A (**c**) and ms^2^t^6^A (**d**) levels in serum was performed for calculation of sensitivity and specificity.

**Figure 5 biomolecules-12-01233-f005:**
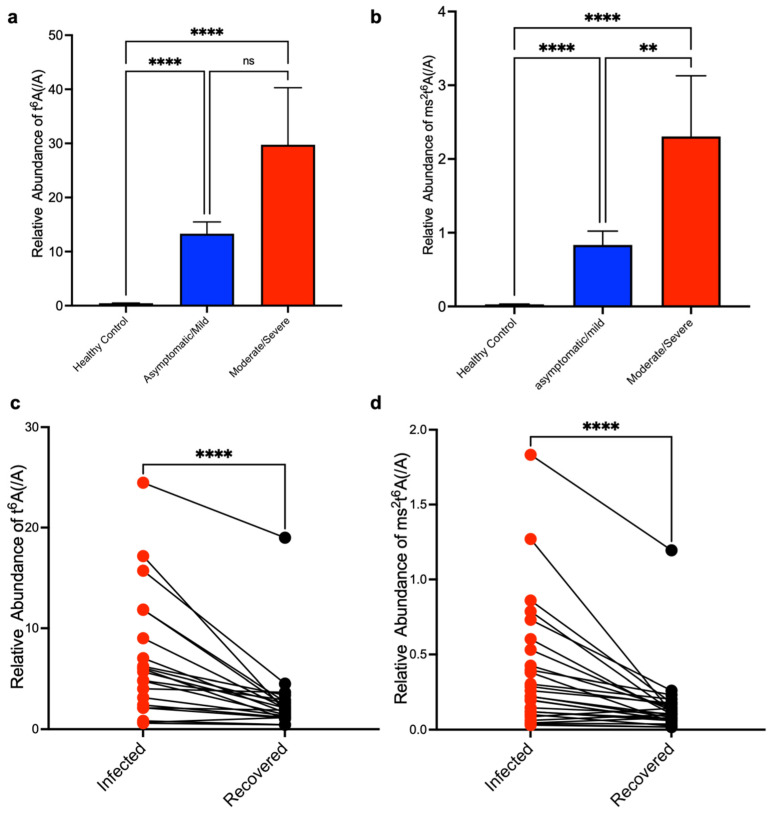
Elevations of t^6^A and ms^2^t^6^A in serum of COVID-19 patients correlate with severity and recovery of COVID-19. (**a**,**b**) Elevation of t^6^A (**a**) and ms^2^t^6^A (**b**) in serum categorized by COVID-19 severity. LC-MS peak areas of t^6^A or ms^2^t^6^A divided by LC-MS peak areas of adenosine are shown. ** *p* < 0.01, **** *p* < 0.0001 by Kruskal–Wallis test and Dunn’s multiple comparison test. (**c**,**d**) Comparison of t^6^A (**c**) and ms^2^t^6^A (d) levels in serum of COVID-19 patients at the time of infection and recovery. A recovery period was defined by the resolution of fever and other symptoms. LC-MS peak areas of t^6^A or ms^2^t^6^A divided by LC-MS peak areas of adenosine are shown. **** *p* < 0.0001 by Wilcoxon rank-sum test.

**Figure 6 biomolecules-12-01233-f006:**
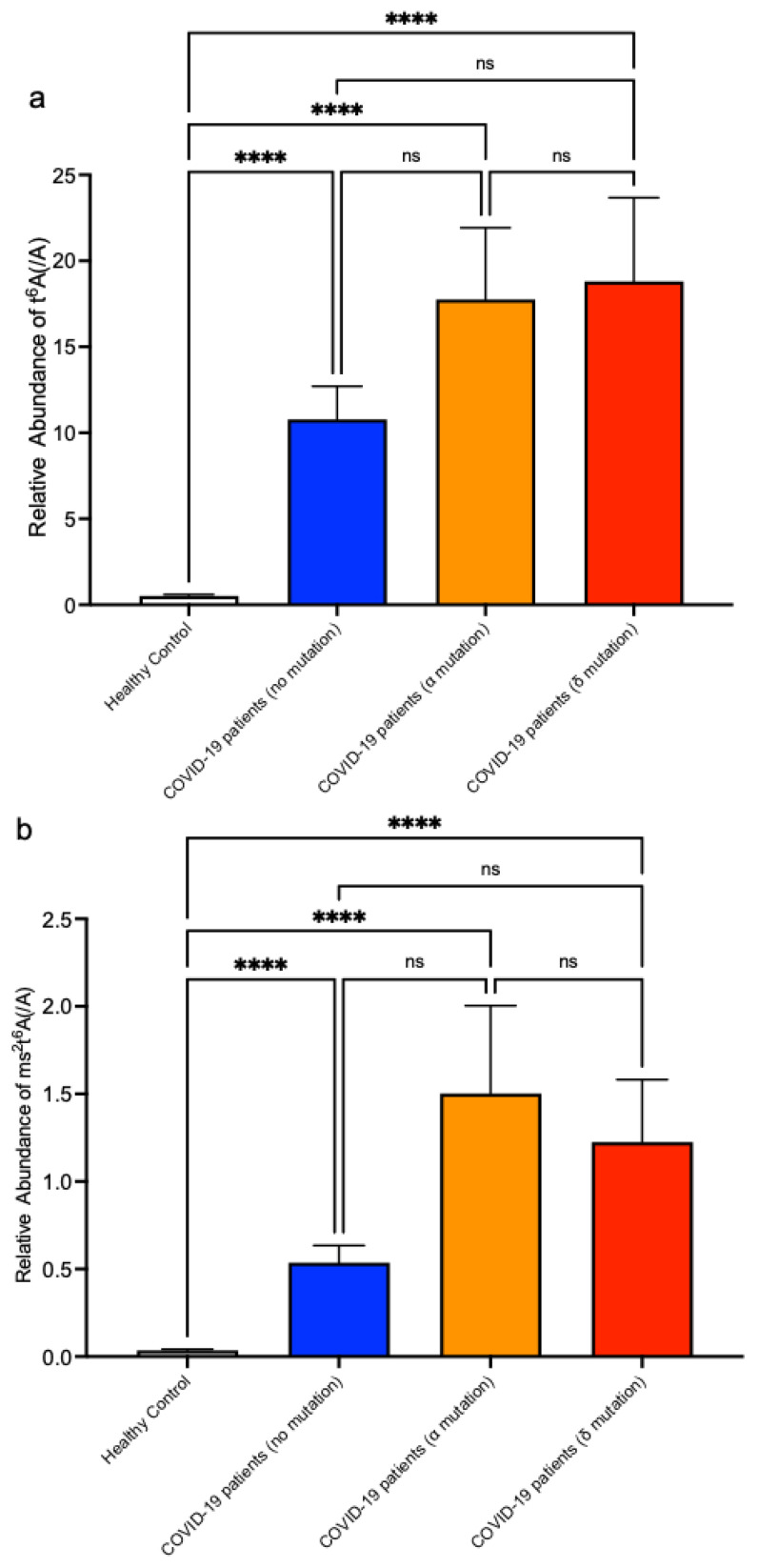
Elevation of t^6^A and ms^2^t^6^A levels in serum of patients infected by different SARS-CoV-2 strains. (**a**,**b**) Measurements of t^6^A (**a**) and ms^2^t^6^A (**b**) in serum of COVID-19 patients infected with α strain or δ strain compared with healthy volunteers. LC-MS peak areas of t^6^A or ms^2^t^6^A divided by LC-MS peak areas of adenosine are shown. **** *p* < 0.0001 by Kruskal–Wallis test and Dunn’s multiple comparison test. ns, not significant.

**Table 1 biomolecules-12-01233-t001:** Patient characteristics of this study.

	Healthy Volunteers (N = 40)	COVID-19 Patients (N = 308)	Bacterial Infection Patients (N = 18)	Viral Infection Patients (N = 24)
Age at inclusion (year 95% CI)	28.1 (31.3–24.9)	50.7 (52.9–48.5)	73.2 (77.9–68.4)	66.7 (71.9–61.4)
Sex				
Male (*n*, %)	24 (60)	178 (57.8)	14 (77.8)	13 (54.1)
Female (*n*, %)	16 (40)	130 (42.2)	4 (22.2)	11 (45.8)
Serum collection	40	308	-	-
Urine collection	10	60	18	24
Race				
East Asian (%)	100	100	100	100
COVID-19 Severity				
Asymptomatic/mild (*n*, %)	-	235 (76.2)	-	-
Moderate/severe (*n*, %)	-	73 (23.8)	-	-
Mutation of SARS-CoV-2				
No mutation (*n*, %)	-	51 (16.6)	-	-
α-mutation (*n*, %)	-	80 (26)	-	-
δ-mutation (*n*, %)	-	177 (57.4)	-	-
CKD patients (eGFR < 60) (*n*, %)	-	69 (20.6)	9 (50)	9 (37.5)
WBC (/μL) (95% CI)	-	5675.1 (5318.7–6033.2)	10.375 (3570–26,980)	6314.1 (3110–9660)
Lymphocyte (%) (95% CI)	-	22.8 (21.3–24.2)		
LDH (U/L) (95% CI)	-	247.8 (232.1–263.5)	-	-
CRP (mg/dL) (95% CI)	-	2.979 (2.41–3.54)	9.17 (0.19–23.02)	1.17 (0.02–3.53)

Abbreviations: eGFR: estimated glomerular filtration rate. CKD: chronic kidney disease. WBC: white blood cells. LDH: lactate dehydrogenase. CRP: C-reactive protein.

## Data Availability

This dataset is available with the permission of the responsible author and the institutional ethics board of Kumamoto University.

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
