# Peer review of "t6A and ms2t6A Modified Nucleosides in Serum and Urine as Strong Candidate Biomarkers of COVID-19 Infection and Severity"

_biomolecules, 2022, doi:10.3390/biom12091233_

Round 1
Reviewer 1 Report
In this manuscript Yu et al. identified two modified nucleosides that were significantly upregulated in the blood and urine of COVID-19 patients and explored their potential as as predictive and typing biomarkers. This is a interesting finding, but the current analysis is preliminary and the content has many uncritical points, the following are some comments.
1. the authors infected HEK293 cells with SARS-CoV-2 and examined the modified nucleic acids in total RNA, and finally identified the two most variable modifications t6A and ms2t6A as the targets of the study. Considering the differences between cell lines and samples, as well as the effects of infection time and concentration, the authors need to supplement the mapping of the changes of various types of modifications in cell lines and fluids under different conditions.
2. The authors listed various clinical indicators of the patients, but did not analyze whether changes in t6A and ms2t6A correlated with these clinical indicators. In addition, the authors need to include more clinical indicators, including various types of immune cell changes including blood routine indicators, and then analyze them item by item.
3. The authors explored the changes of t6A and ms2t6A in blood and urine of COVID-19 patients, respectively, and analyzed their effects as predictive and staging markers. However, the authors need to analyze the two factors jointly to build a classification model and I believe that better results will be obtained.
4. Although the overall trend of infection was consistent among different SARS-CoV-2 mutant strains, were there significant differences in the changes among the mutant strains?
5. The authors tried to show that t6A and ms2t6A can be used as markers, however, without validation from multicenter or external datasets, this work can only show the potential of having biomarkers. The authors either need to add more data or revise the title of the article.
6. The authors mention in the discussion that the changes in t6A and ms2t6A may be due to changes in immune cells, which contradicts the cell line experiments mentioned in the opening paragraph and from the current evidence can only indicate a tissue cell origin.
7. t6A and ms2t6A are modifications specific to two tRNA molecules, and SARS-CoV-2 virus does not encode tRNA in the genome; therefore, it is unlikely to be a molecule of viral origin, but more likely to be caused by an increase in tRNA molecules or degradates in blood or urine, and the authors should cite relevant evidence to increase the conviction of the discussion.
Reviewer 2 Report
The authors discovered serum and urine t6A and ms2t6A nucleosides as effective biomarkers of COVID-19. A mass spec method using triple-Quad-MS in MRM/SRM mode was developed to detect the different nucleosides.
Some comments:
1 1) For bioassyays, the two parameters specificity and sensitivity are usually given. Could you calculate these please?
22) The authors claim in line 297:” In the COVID-19 treatment, RT-PCR tests and antigen tests are not suitable for the proper assessment of COVID-19 severity.” Could that be discussed a little more critically?
33) Why should ELISAtest less quantitative for proteins than for small molecules?
44) Why weren't the nucleosides quantified? The ratio to another naturally occurring nucleoside is too erroneous.
55) Why was no internal standard used?
66) What kind of false positive values do you get for cancer patients?
Reviewer 3 Report
The work of Nagayoshi et al. describes the intriguing observation that HEK293 cells infected with sars-cov2 produce higher amounts of t6A and m2st6A modified nucleosides. Importantly, these same metabolites are present in serum and urine samples from COVID19 patients in higher amounts than in control samples or samples from patients suffering other viral or bacterial respiratory diseases. It would be important to include as supplementary data the diagnosis in each case.
Additionally, the detection of these modified nucleosides is safer than PCR or other methods based on the presence of viral particles. However, in my opinion the most outstanding finding is the fact that the amount of modified nucleoside detected correlates with the severity of the symptoms developed by the patient, this is especially useful for the management of resources in any health system. Also, the quantity of t6A and ms2t6A show the remission of the infection and is a good biomarker for the different sars-cov2 variants.
The description of sample preparation for LC-MS analysis well explained. This technique is very sensitive allowing the detection of these metabolites in a small fluid sample.
From the basic scientific perspective, the manuscript gives little information about what is known regarding the synthesis and presence of this modified nucleosides in nucleic acids. This should be included in the introduction.
Cdkal1 is responsible for the canonical synthesis of ms2t6A specifically in UUU decoding tRNAs. This should be discussed as well as the different context that could modify the amount of the subunits of the KEOPS complex (responsible for t6A synthesis in ANN decoding tRNAs) or their activity.
Authors, based on direct nanopore sequencing of viral genome, suggests that t6A and m2st6A could be modifications present in sars-cov2 genome however these modifications have never been detected in viral RNAs. The source of plasma nucleoside could be related with immune cells and the cytokine storm produced in COVID19 patients and potential cell death generated under these circumstances.
In the analysis of modified nucleosides in serum samples it is plotted the abundance of each metabolite normalized by total Adenosine. If these metabolites result from RNA molecules it is very intriguing that this ratio (modified nucleoside/Adenosine) is so large. This should be considered in the discussion section.
Since t6A/A ratio present no significant difference between severity groups (Fig.5 a), it should not be claimed that t6A detection in serum samples reveal severity and this statement only apply for ms2t6A.
In this section it would be useful to know how recovery is defined, how many days after a negative PCR the sample was taken?
Round 2
Reviewer 1 Report
The Authors have addressed all of my concerns.